# Oscillatory movement of a dynein-microtubule complex crosslinked with DNA origami

Shimaa A Abdellatef[1,2], Hisashi Tadakuma[3,4], Kangmin Yan[1], Takashi Fujiwara[5], Kodai Fukumoto[3], Yuichi Kondo[5], Hiroko Takazaki[3,6], Rofia Boudria[1,7], Takuo Yasunaga[6], Hideo Higuchi[5], Keiko Hirose[1]*

[1]Biomedical Research Institute, National Institute of Advanced Industrial Science and Technology (AIST), Tsukuba, Japan; [2]Research Center for Functional Materials, National Institute for Materials Science, Tsukuba, Japan; [3]Institute for Protein Research, Osaka University, Osaka, Japan; [4]SLST and Gene Editing Center, ShanghaiTech University, Shanghai, China; [5]Graduate School of Science, The University of Tokyo, Tokyo, Japan; [6]Kyushu Institute of Technology, Fukuoka, Japan; [7]Institut Pasteur, Paris, France

**Abstract** Bending of cilia and flagella occurs when axonemal dynein molecules on one side of the axoneme produce force and move toward the microtubule (MT) minus end. These dyneins are then pulled back when the axoneme bends in the other direction, meaning oscillatory back and forth movement of dynein during repetitive bending of cilia/flagella. There are various factors that may regulate the dynein activity, e.g. the nexin-dynein regulatory complex, radial spokes, and central apparatus. In order to understand the basic mechanism of dynein's oscillatory movement, we constructed a simple model system composed of MTs, outer-arm dyneins, and crosslinks between the MTs made of DNA origami. Electron microscopy (EM) showed pairs of parallel MTs crossbridged by patches of regularly arranged dynein molecules bound in two different orientations, depending on which of the MTs their tails bind to. The oppositely oriented dyneins are expected to produce opposing forces when the pair of MTs have the same polarity. Optical trapping experiments showed that the dynein-MT-DNA-origami complex actually oscillates back and forth after photolysis of caged ATP. Intriguingly, the complex, when held at one end, showed repetitive bending motions. The results show that a simple system composed of ensembles of oppositely oriented dyneins, MTs, and inter-MT crosslinkers, without any additional regulatory structures, has an intrinsic ability to cause oscillation and repetitive bending motions.

*For correspondence: k.hirose@aist.go.jp

**Competing interest:** The authors declare that no competing interests exist.

## Editor's evaluation

The authors describe the reconstitution of axonemal bending using polymerized microtubules, purified outer-arm dyneins, and synthesized DNA origami to cross-link two microtubules. The work is of interest to the field as it shows that bidirectional sliding and bending of microtubules can be generated by a minimal set of elements.

## Introduction

Beating of cilia and flagella is powered by axonemal dynein molecules that are minus-end-directed microtubule (MT) motors (*Gibbons and Rowe, 1965*). Each dynein is tethered to a doublet MT with its tail, and interacts with the neighboring MT using its stalks in an ATP-dependent way, to produce

**Figure 1.** Design of the dynein-microtubule-DNA-origami (dynein-MT-DNA-origami) complex. (**A**) Schematic models showing directions of dynein sliding and bending of the axoneme. Dynein molecules between two pairs of doublet MTs, one pair on the far-side (top) and another pair on the near-side (bottom) are illustrated. When viewed from the front of the axoneme, the stalk and head of a far-side dynein are seen to the left of the tail, whereas those of a near-side dynein are on the right. As a result, the axoneme bends in opposite directions when only the far-side or only the near-side dyneins are activated and move toward the minus ends (colored in orange). Bending of the axoneme causes sliding of dynein on the unactivated side (yellow) toward the plus end. (**B**) Geometry of the dynein-MT complex. Longitudinal and axial views are illustrated. *Chlamydomonas* outer-arm dynein molecules crossbridge two MTs, binding with their tails and stalks. The stalks point toward the MT minus end. Because the tails and stalks bind to different MTs, there are two possible orientations for dynein depending on which MT the stalks bind to. (**C**) Schematic illustration of a cross-sectional view of an axoneme. Note that the geometry of dyneins in the dynein-MT complex shown in (**B**) mimics that of a combination of the dyneins on two opposite sides of the axoneme (cyan boxes), although the dynein arrays in (**B**) are not continuous. (**D**) Design of rod-shaped DNA origami. For binding to MTs, mutant kinesin motor domains (E237A) were attached to 5'-SNAP-ligand-modified handles placed on the DNA rod. The two handles were separated by ~50 nm, corresponding to the center-to-center distance of the two MTs in the dynein-MT complex. Each handle has 30 nucleotides of poly-thymidine linker between the SNAP ligand and the rod. (**E**) Geometry of the dynein-MT complex crosslinked with DNA-origami rods. In the presence of ATP, dynein molecules bound in the opposite orientations are thought to produce force in opposite directions (indicated by arrows). DNA rods are expected to crosslink the two MTs of the complex and restrict their relative movement.

relative sliding of the two MTs. The MTs in an axoneme are all parallel and arranged in a 9 + 2 structure with their minus ends at the base and interconnected with various components, such as the nexin-dynein regulatory complex (*Heuser et al., 2009*). Because the neighboring doublet MTs are inter-linked and also anchored to the basal body at the base, minus-end-directed movement of the dynein molecules on one side of the axoneme causes bending of the axoneme in one direction (*Figure 1A*). These dynein molecules are then pulled back (i.e. toward the plus end of the MT) when the dyneins on the opposite side of the axoneme produce force to bend the axoneme in the other direction. Thus, during cyclical bending of the axoneme, dynein molecules oscillate back and forth along a MT. How a minus-end-directed motor dynein can move back and forth along a MT is unknown.

There are many factors that may influence force production by dynein. For example, outer-arm dyneins and several species of inner-arm dyneins are thought to affect each other's activities. The arrangement of the dyneins in an axoneme must be also important for their communication; the outer-arm dyneins are aligned in a row with a periodicity of 24 nm, and the inner-arm dyneins are also regularly placed in specific positions (*Bui et al., 2008*). The inter-doublet linkers connect the circumferentially arranged nine doublet MTs, and thus transmit the force produced locally by some dyneins to the other parts of the axoneme (*Lindemann, 2003*). Other regulatory components, such as the radial spokes and central apparatus also seem to play important roles in the regulation (*Witman et al., 1978*). In order to minimize complexity and elucidate only the basic mechanism of cilia and flagella motility, we designed a simple model system and tested its structure and motility. Our system consists of in vitro polymerized MTs, axonemal outer-arm dynein molecules, and passive linkers that interconnect pairs of parallel MTs (*Figure 1E*). This simple model system does not have the circular arrangement of nine MTs in the axoneme, nor does it have inner-arm dyneins, radial spokes or the central apparatus.

To reconstitute the regular arrangement of outer-arm dyneins, which allow neighboring dynein molecules to interact with each other, we used dynein preparations extracted from *Chlamydomonas* flagella axonemes with high salt (typically 0.6 M KCl). The high-salt-extracted dynein preparations are known to crossbridge MTs with ~24 nm periodicity (*Haimo et al., 1979*), and although they contain other components in addition to outer-arm dyneins, previous work showed that the proteins bound to MTs are mostly outer-arm dynein (*Aoyama and Kamiya, 2010*; *Oda et al., 2007*). As inter-MT linkers, we utilized rod-shaped DNA origami structures and attached the DNA rods to the MTs via flexible linkers (*Figure 1D and E*). Use of DNA origami enabled us to design a molecular layout of the cross-linking structure that allows relative sliding of the MTs over a certain distance, resulting in oscillation and repetitive bending motions as seen in intact axonemes.

## Results

### Geometry of dynein-MT complexes

Dynein preparations extracted with high salt from *Chlamydomonas* flagella axonemes were mixed with taxol-stabilized MTs polymerized from brain tubulin (*Figure 1B*). As previously reported (*Aoyama and Kamiya, 2010*; *Haimo et al., 1979*; *Oda et al., 2007*), the MTs became bundled with the cross-bridging molecules bound with a period of ~24 nm (*Figure 2A and B*). In the dynein-MT complexes prepared with high concentrations of dynein, a pair of MTs in bundles are crossbridged by two continuous arrays of dynein, so that superposition of two rows of dynein molecules is observed in electron microscopy (EM) images (*Haimo et al., 1979*; *Oda et al., 2007*). On the other hand, when a low concentration of the dynein preparation (6.25–12.5 μg/ml [corresponding to ~3–6 nM outer-arm dynein]) was mixed with 20–25 μg/ml MTs (200–250 nM tubulin dimers), the MTs were only partially decorated with dynein, so that we were able to observe single layers of crossbridges without superposition in many regions. Negative-stain EM observation of the individual crossbridges showed characteristic shapes of outer-arm dynein, with stacked heads and a tail (*Heuser et al., 2009*; *Movassagh et al., 2010*; *Figure 2C*), confirming that the crossbridging proteins are mostly outer-arm dyneins. The average number of dyneins per 1 μm of a MT pair was about 12 when 6.25 μg/ml dynein was used.

When a dynein molecule crossbridges a pair of MTs, its tail is fixed onto one of the MTs and the stalks interact with the other MT in a nucleotide-dependent manner. Therefore, depending on which MT the tail binds to, there are two possible orientations (*Figure 1B*). EM images confirmed that dyneins crossbridge MTs in two different orientations (*Figure 2A and B*). As observed previously with sea urchin dynein (*Hirose, 2012*; *Ueno et al., 2008*), the crossbridging dyneins made patches, and the adjacent dynein molecules within one patch tended to have the same orientation. In vivo, outer-arm dynein binds to MTs with its head + stalk oriented toward the minus end of the MTs, and its tail oriented toward the plus end (*Goodenough and Heuser, 1982*; *Heuser et al., 2009*; *Movassagh et al., 2010*), so that the two MTs crosslinked with dyneins have the same polarity.

We have previously shown by cryo-EM analysis that in vitro also, dynein's head and stalk are always oriented toward the minus end of the MT to which its stalk binds (*Ueno et al., 2008*). Thus, we determined the polarity of the MTs in the dynein-MT complex based on the orientation of dyneins in the EM images. Whereas the orientation of the head + stalk is always toward the minus end, binding of

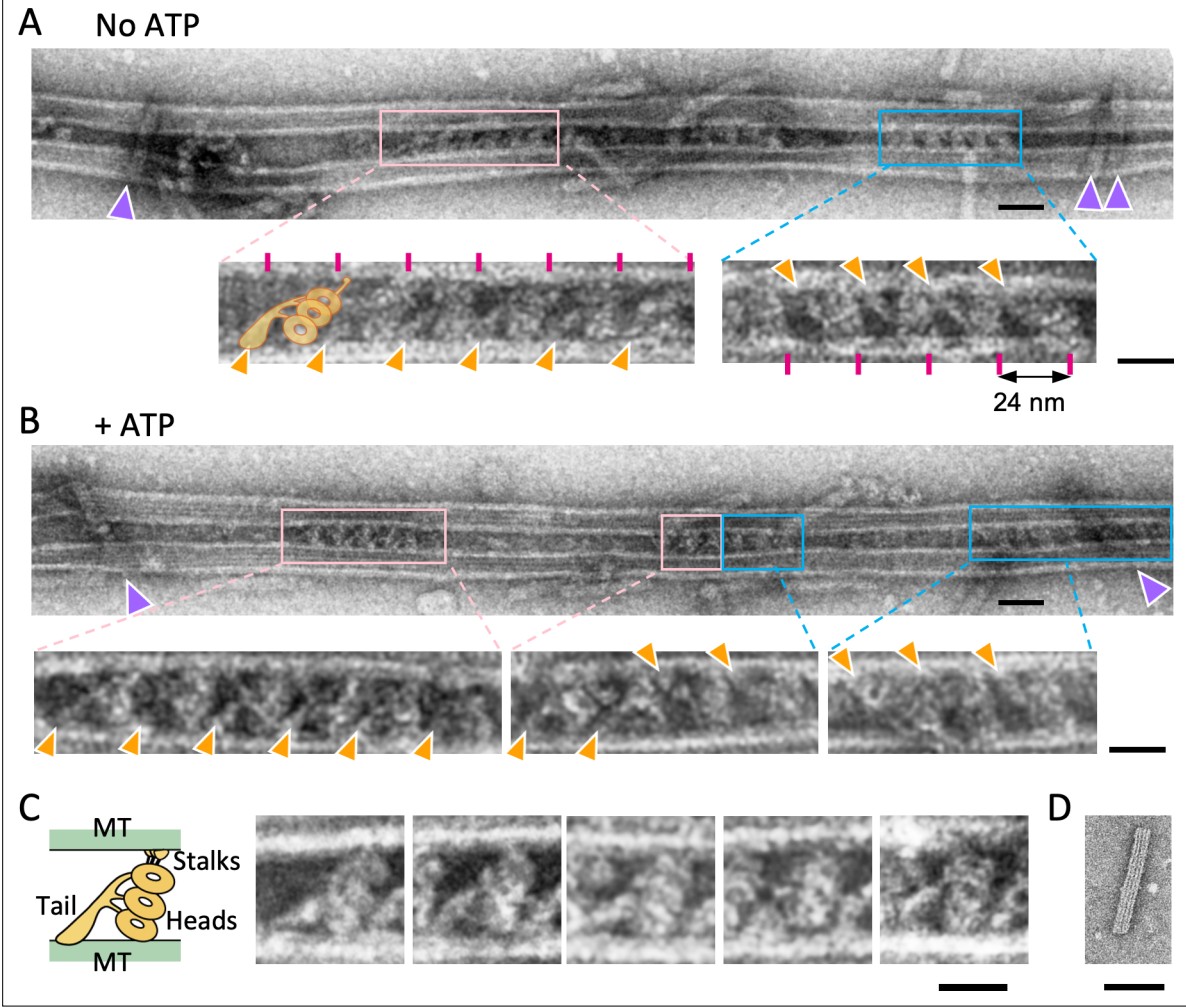

**Figure 2.** Structure of the dynein-microtubules-DNA-origami (dynein-MT-DNA-origami) complex. (**A–B**) Negative stain images of the dynein-MT-DNA-origami complex in the absence (**A**) and presence (**B**) of 0.1 mM ATP. DNA rods crosslinking the MTs are indicated by purple arrowheads. Dynein molecules tend to bind in patches and the neighboring dyneins usually have the same orientation. Patches of dynein cross-bridging the MTs in two different orientations are indicated by pink and cyan boxes. Enlarged views show examples of dyneins in two orientations, with their tails (indicated by orange arrowheads) on different MTs. Magenta lines show a periodicity of 24 nm. The typical shape of an outer-arm dynein is illustrated. Bars: 50 nm in low magnification images and 20 nm in enlarged views. (**C**) Images of individual dynein molecules crossbridging two MTs. As illustrated on the left, the characteristic shapes of a *Chlamydomonas* outer-arm dynein molecule (*Heuser et al., 2009*; *Movassagh et al., 2010*) with a tail and three heads are observed. Bar: 20 nm. (**D**) A negative stain image of a DNA origami rod. Bar: 50 nm.

The online version of this article includes the following source data and figure supplement(s) for figure 2:

**Source data 1.** Numerical data for *Figure 2—figure supplement 4*.

**Figure supplement 1.** Geometries of the dynein-microtubule (dynein-MT) complex.

**Figure supplement 2.** A negative stain image of DNA origami rods.

**Figure supplement 3.** Structures of dynein molecules in the absence and presence of ATP.

**Figure supplement 4.** Apparent lengths of microtubule (MT)-bound dynein molecules in the absence and presence of ATP measured in negative stain images.

the tail to the MT was less specific in vitro, so that the resulting complexes had either parallel or anti-parallel MTs (*Ueno et al., 2008*) as illustrated in *Figure 2—figure supplement 1A, B*. In the case of sea urchin dynein, the two MTs of a complex were often antiparallel (*Ueno et al., 2008*; *Yokota and Mabuchi, 1994*). In contrast, the majority of the *Chlamydomonas*-dynein-MT complexes (36 out of 44 complexes) showed parallel arrangements of the MTs as in axonemes (*Figure 2—figure supplement 1C, D*).

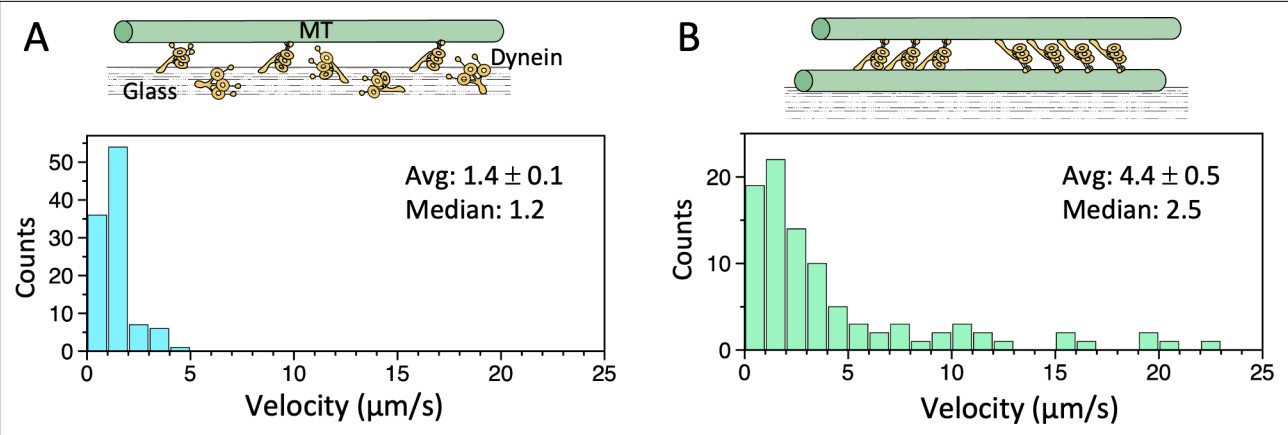

**Figure 3.** Microtubule (MT)-gliding velocities in two different conditions. (**A**) Distributions of the MT gliding velocities in usual MT gliding assays in which MTs move over dynein-coated glass surfaces. (**B**) Relative sliding velocities of the MTs in a dynein-MT complex that contains dyneins in two opposite orientations. Average (mean ± SEM) and median values of the velocities are indicated (n=104 and 94 for A and B, respectively). Also see *Figure 3—source data 1*.

The online version of this article includes the following source data and figure supplement(s) for figure 3:

**Source data 1.** Numerical data for *Figure 3*.

**Figure supplement 1.** Maximum force produced by the dynein-microtubule (dynein-MT) complex without DNA origami rods.

In the complex with parallel MTs, the dynein molecules in two opposite orientations are expected to produce force in opposite directions (*Figure 1B* and *Figure 2—figure supplement 1A*), as in the models previously proposed (*Camalet and Jülicher, 2000*; *Mitchison and Mitchison, 2010*; *Riedel-Kruse et al., 2007*). Compared to the continuous dynein arrays in axonemes, the arrays of dynein in our complex were much shorter and discontinuous because we used low concentrations of dynein. Nevertheless, the arrangement of dyneins in our complex is analogous to a combination of two dynein arrays on the opposite sides of an axoneme, in the sense that both systems contain two subsets of regularly arranged dynein molecules, and the subsets produce antagonistic forces (compare the axial views in *Figure 1B and C*). The resemblance in their geometries led us to the idea that these dynein-MT complexes might work as a simple model system to investigate the minimum components required for oscillatory movements that occur in cilia and flagella.

## Motile properties of the dynein-MT complex

The outer-arm dynein preparations used here supported movement of MTs in gliding assays at a speed comparable to those in previous work (*Alper et al., 2013*; *Furuta et al., 2009*; *Figure 3A*, *Figure 3—source data 1*). We then tested the motility of dynein-MT complexes that contain a pair of MTs crossbridged by dyneins in two different orientations (*Figure 3B*). When the dynein-MT complexes are adsorbed to a

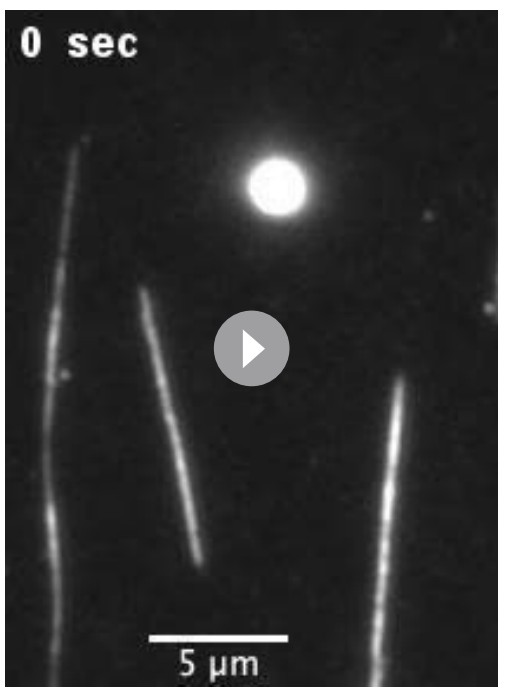

**Video 1.** Unidirectional movement of microtubules (MTs) from the dynein-MT complex. Dynein-MT complexes were adsorbed to the glass surface. Upon photolysis of caged ATP, some MTs (indicated by arrowheads) moved unidirectionally and slid out of the bundles. Recorded at 400 ms/frame.

https://elifesciences.org/articles/76357/figures#video1

glass surface, the MTs that are directly attached to the glass cannot move, but other MTs in the same complex are allowed to slide relative to the glass-attached MTs. Relative sliding of the MTs leads to disassembly of the complex as soon as the complexes are exposed to an ATP-containing solution. Therefore we used caged ATP and observed the movement of MTs immediately after the ATP was uncaged locally by photolysis. Even though the majority of the complexes are thought to contain groups of dynein molecules that produce force in opposite directions, many of the complexes showed unidirectional sliding of MTs (*Video 1*), probably because the numbers of molecules in the two groups are not equal and the dominant group would 'win'. The velocity was variable (*Figure 3B*, *Figure 3—source data 1*), with some MTs sliding much faster than those observed in gliding assays. Fast sliding was previously observed with similar dynein-MT complexes but with much higher concentration of dynein, and was interpreted as the effect of cooperation of dyneins aligned with 24 nm periodicity (*Aoyama and Kamiya, 2010*).

We have also measured the force produced by the dynein-MT complexes using optical trapping methods (*Figure 3—figure supplement 1*). The average number of dyneins that crossbridge the pair of MTs was estimated to be ~35 from inspection of the EM images (12 molecules/μm) and the average length of the MT (2.9 ± 0.9 μm [mean ± SD]). Typical traces showed a maximum force of ~30 pN. The value is several times larger than the force produced by single molecules of outer-arm dynein from sea urchin (5–6 pN) (*Shingyoji et al., 2015*) or *Tetrahymena* (4.7 pN) (*Hirakawa et al., 2000*), indicating cooperative force production. On the other hand, it is smaller than the simple sum of the forces produced by each dynein. Dependence of the maximum force on the number of motors differs in different motors (*Furuta et al., 2013*; *Soppina et al., 2009*). The fact that the dependence observed here is not proportional to the dynein number may be related to the weak processivity of the molecule. Alternatively, the dynein molecules oriented in the opposite direction may act as a load.

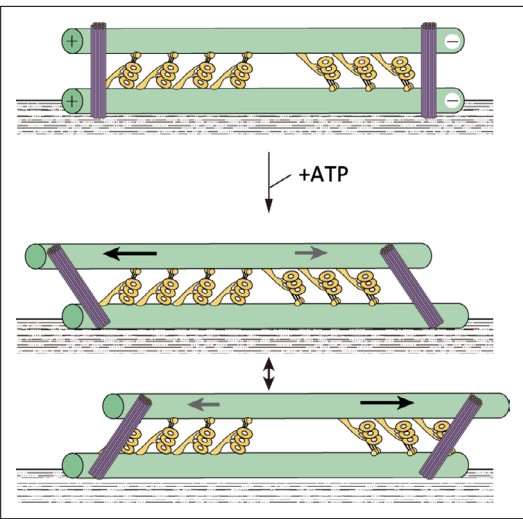

**Figure 4.** A model showing oscillatory movement of a dynein-microtubule-DNA-origami (dynein-MT-DNA-origami) complex. The two groups of dyneins produce opposing forces. DNA rods are expected to crosslink the two MTs of the complex and restrict their relative movement. When the total force produced by one of the groups is stronger, the upper MT moves in one direction and the DNA rods tilt. Movement of the MT would stop when the linkers between the MTs and DNA rods are fully extended. The MT would then move in the opposite direction until the movement is again stopped by the DNA rods.

The online version of this article includes the following video, source data, and figure supplement(s) for figure 4:

**Source data 1.** Numerical data for *Figure 4—figure supplement 2* (tilt angles of DNA origami rods).

**Source data 2.** Numerical data for *Figure 4—figure supplement 2* (Mann-Whitney U-test).

**Figure supplement 1.** Estimation of tilt angles of DNA origami rods.

**Figure supplement 2.** Distributions of the tilt angles measured in the negative stain electron microscopy (EM) images.

**Figure 4—video 1.** Effect of DNA-origami rods on the movement of dynein-microtubule (dynein-MT) complexes.

https://elifesciences.org/articles/76357/figures#fig4video1

## Crosslinking of the dynein-MT complex with DNA origami structures

The above results showed that a dynein-MT complex alone is not sufficient to generate oscillatory movement, even though it contains two groups of dyneins that produce force in opposite directions: the movement was mostly in one direction and the complex disassembled. To prevent disassembly and mimic axonemal doublet MTs that are interconnected with linkers, we crosslinked the MTs of the dynein-MT complex. A rod-shaped DNA origami of ~84 nm in length was attached to the dynein-MT complex via immotile mutant kinesin motor domains (E237A) (*Rice et al., 1999*; *Figure 1D and E*,

and *Figure 2—figure supplement 2*). In vitro motility assays confirmed that addition of enough DNA origami rods (e.g. 2.5 nM DNA rods for the dynein-MT complex with 6.25 µg/ml dynein) can actually prevent disassembly of the complex in the presence of ATP, so that movement of MTs was not detected by fluorescence microscopy (*Figure 4—video 1*).

In order to allow relative sliding of the MTs and possible oscillation in the presence of ATP, flexible poly-T linkers were inserted between the DNA rod and kinesin (*Figure 4*, *Figure 4—figure supplement 1A*). The maximum relative sliding distance was estimated to be ~96 nm in both directions in the ideal situation where all the DNA rods are bound perpendicular to the MTs. The actual sliding distance is expected to be shorter because the initial binding of the DNA rods is not necessarily perpendicular (*Figure 4—figure supplement 1B*).

DNA origami rods crosslinking the dynein-MT complex were clearly visible by EM (*Figure 2A and B*, indicated by purple arrowheads; *Figure 2D*). The average number of DNA rods crosslinking a pair of MTs was ~2.4 per µm when 2.5 nM DNA rods were used. As expected, the binding angle in the absence of ATP was variable but centered around the perpendicular (*Figure 4—source data 1*,

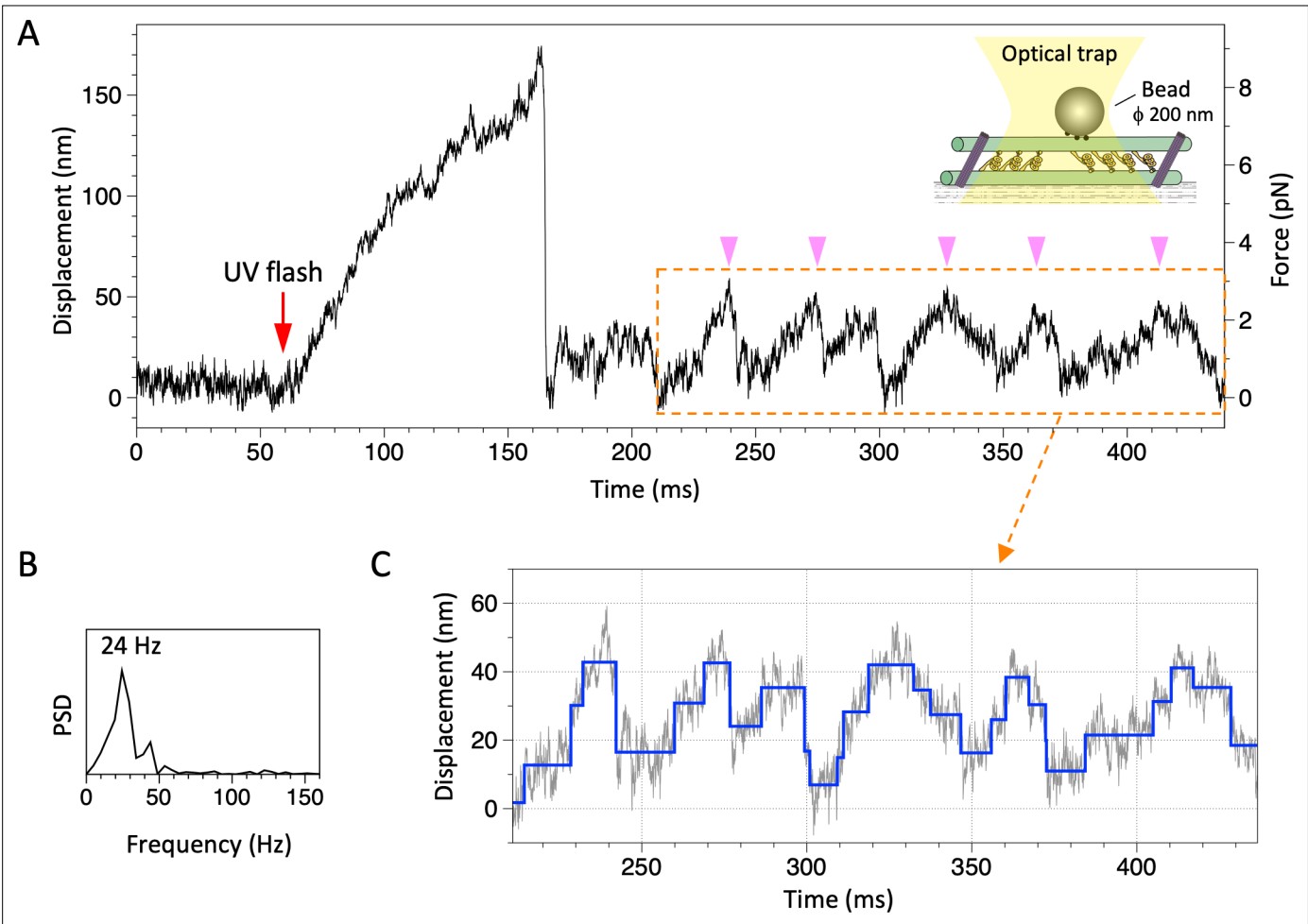

**Figure 5.** Movement of the dynein-microtubule-DNA-origami (dynein-MT-DNA-origami) complex measured in optical trapping assays. (**A**) Trace showing displacement of a bead attached to the dynein-MT-DNA-origami complex after UV photolysis of caged ATP (red arrow). Schematic of the experimental set-up is shown in inset (not to scale). A streptavidin-coated bead was captured by the optical trap (trap stiffness 0.052 pN/nm) and attached to a dynein-MT-DNA-origami complex adsorbed to a glass surface. A part of the trace shows oscillatory movement (pink arrowheads). (**B**) The frequency of the oscillatory movement was measured by the power spectral density (PSD). (**C**) Steps (blue) detected for the same region by the step-finding algorithm (*Kerssemakers et al., 2006*). Steps are found in both forward (away from the trap center) and backward (toward the trap center) movements.

The online version of this article includes the following figure supplement(s) for figure 5:

**Figure supplement 1.** Displacement of beads attached to the dynein-microtubule-DNA-origami (dynein-MT-DNA-origami) complex.

*Figure 4—figure supplement 2*). When ATP was added to the complex, the deviation of the rod angle from the perpendicular increased (p<0.002, Mann-Whitney U-test; *Figure 4—source data 2*), indicating that the DNA rods tilted because of relative sliding of the MTs.

## Dynein-MT-DNA-origami complexes show oscillatory movements

We then investigated the motility of the dynein-MT-DNA-origami complexes in detail. Since the expected relative sliding of the MTs was too small to be detected by fluorescence microscopy, we attached a bead to the complex and measured the displacement after photolysis of caged ATP (*Figure 5*). In some cases, the bead bound to a MT moved unidirectionally and then abruptly went back to the trap center, probably because of detachment of dynein from the MT (e.g. the first ~100 ms after the UV flash in *Figure 5A*), as observed in previous work using single dynein molecules (e.g. *Hirakawa et al., 2000*; *Sakakibara et al., 1999*; *Toba et al., 2006*).

The most notable feature observed in the optical trapping measurement of the dynein-MT-DNA-origami complex was oscillatory movements. Typical traces are shown in *Figure 5A* and *Figure 5—figure supplement 1*. Although the oscillation was irregular, it was clearly different from the noisy vibration before the UV flash (compare the trace before the UV flash and the region boxed in orange in *Figure 5A*), and the power spectrum density showed clear peaks (*Figure 5B* and *Figure 5—figure supplement 1*, right panels), which were not observed before the UV flash. Since caged ATP was locally photolyzed using a UV spot with a full width at half maximum of ~20 µm and the released ATP diffuses in the solution, we analyzed the traces typically within ~0.5 s after photolysis of ATP. About 48 out of 94 such traces showed sliding movement, and 65% of them exhibited oscillatory movement (at least two forward and two backward displacements, each displacement larger than 10 nm, and the velocity of each movement between 0.1 and 50 µm/s) in some parts of the trace. The amplitude of the oscillatory movements was variable, but typically within the expected maximum displacement (~96 × 2 nm, see above), with the averages of 26.6 and 26.4 nm for the forward (away from the trap center) and backward (back toward the trap center) movement, respectively (*Figure 6A and B* and *Figure 6—source data 1*). The velocities of the forward and backward movements during oscillation were also variable (*Figure 6A and D* and *Figure 6—source data 1*), but they were both in the same range as the velocity observed in the gliding assay of the dynein-MT complex without DNA rods (*Figure 3B*; 4.4 µm/s in average). The difference between the averaged velocities of the forward and backward movements (3.8 and 6.4 µm/s, respectively) may be because the force of the optical trap works as a load for the forward movement, whereas it assists the backward movement. Some detachment may also contribute to the faster velocity of the backward movement. However, the average time required for the forward and backward movements (10.1 and 8.3 ms, respectively; *Figure 6A and C* and *Figure 6—source data 1*) was substantially longer than the time required for simple detachment from a MT, which was usually less than 1 ms. The frequencies of the oscillation were measured using the power spectrum of the traces (*Figure 5B* and *Figure 5—figure supplement 1*, right panels). The frequencies were variable, with an average of 32.9 Hz (*Figure 6E*, *Figure 6—source data 1*).

Although many of the traces were noisy, a step-finding algorithm (*Kerssemakers et al., 2006*) detected stepwise movements in both the unidirectional and oscillatory movements of some traces (*Figure 5C*, *Figure 5—figure supplement 1A*). Previous work using single *Tetrahymena* dynein molecules reported steps of ~8 nm at an extreme condition: very low concentration of ATP (3 µM) (*Hirakawa et al., 2000*). Cytoplasmic dynein also showed predominantly 8 nm steps (*Reck-Peterson et al., 2006*; *Toba et al., 2006*), but the step size depended on load (*Belyy et al., 2014*; *Gennerich et al., 2007*). Although many of the traces of our dynein-MT-DNA-origami complex were noisy and we could not accurately measure the step sizes, the peaks of the step sizes were close to 8 nm (6.2, 8.4, and 9.5 nm for unidirectional movement, forward, and backward displacements of the oscillatory movement, respectively; *Figure 6F*, *Figure 6—source data 1*). If the oscillatory movements we observed are simply the cycles of unidirectional movement and dissociation of dyneins, we would not detect multiple steps during the backward movements. Thus, the results indicate that both forward and backward displacements during the oscillatory movement are steps of dynein.

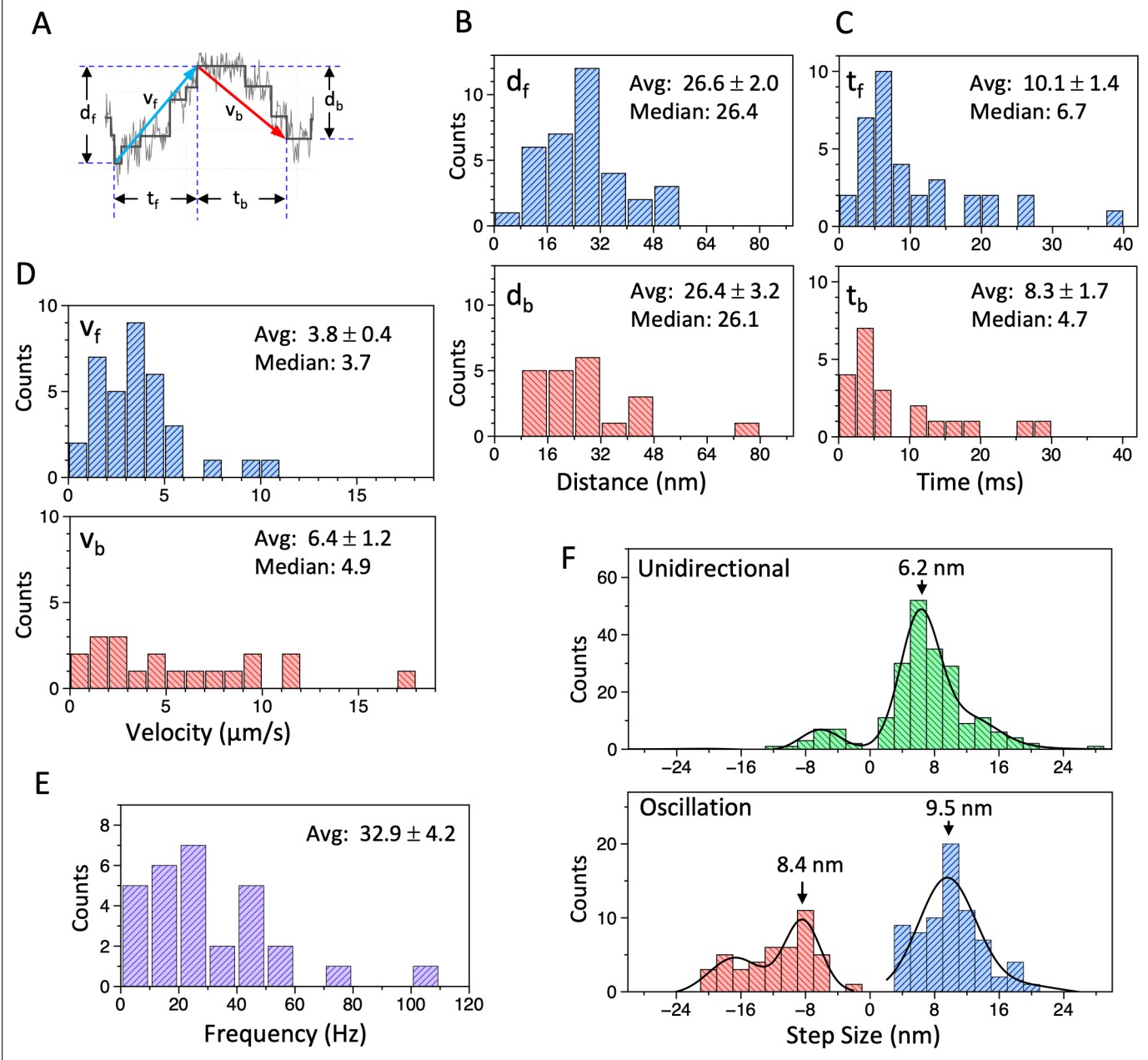

**Figure 6.** Analysis of movement of the dynein-microtubule-DNA-origami (dynein-MT-DNA-origami) complex measured in optical trapping assays. (**A**) Definition of the distance ($d_f$, $d_b$), time ($t_f$, $t_b$), and velocity ($v_f$, $v_b$) used for the histograms in (B – D). (**B – D**) Histograms of the distance (**B**), time (**C**), and velocity (**D**) during the forward ($d_f$, $t_f$, $v_f$) and backward ($d_b$, $t_b$, $v_b$) movements of the bead attached to the dynein-MT-DNA-origami complex. Average (mean ± SEM) and median values are indicated in each histogram (n=35 for $d_f$, $t_f$, $v_f$, and 21 for $d_b$, $t_b$, $d_b$). (**E**) A histogram of the frequency of the oscillatory movements (n=29). (**F**) Histograms of the step size during unidirectional movement (top; n=211) and oscillatory movement (bottom; n=72 for forward and n=44 for backward), fit with multiple Gaussian curves. Main peak positions of the Gaussian functions are indicated. Also see *Figure 6— source data 1*.

The online version of this article includes the following source data for figure 6:

**Source data 1.** Numerical data for *Figure 6*.

## An ensemble of uniformly oriented dynein molecules moves a MT unidirectionally

The above results show that a dynein-MT complex containing two groups of oppositely oriented dyneins can move in an oscillatory manner when the MTs are connected with DNA origami linkers,

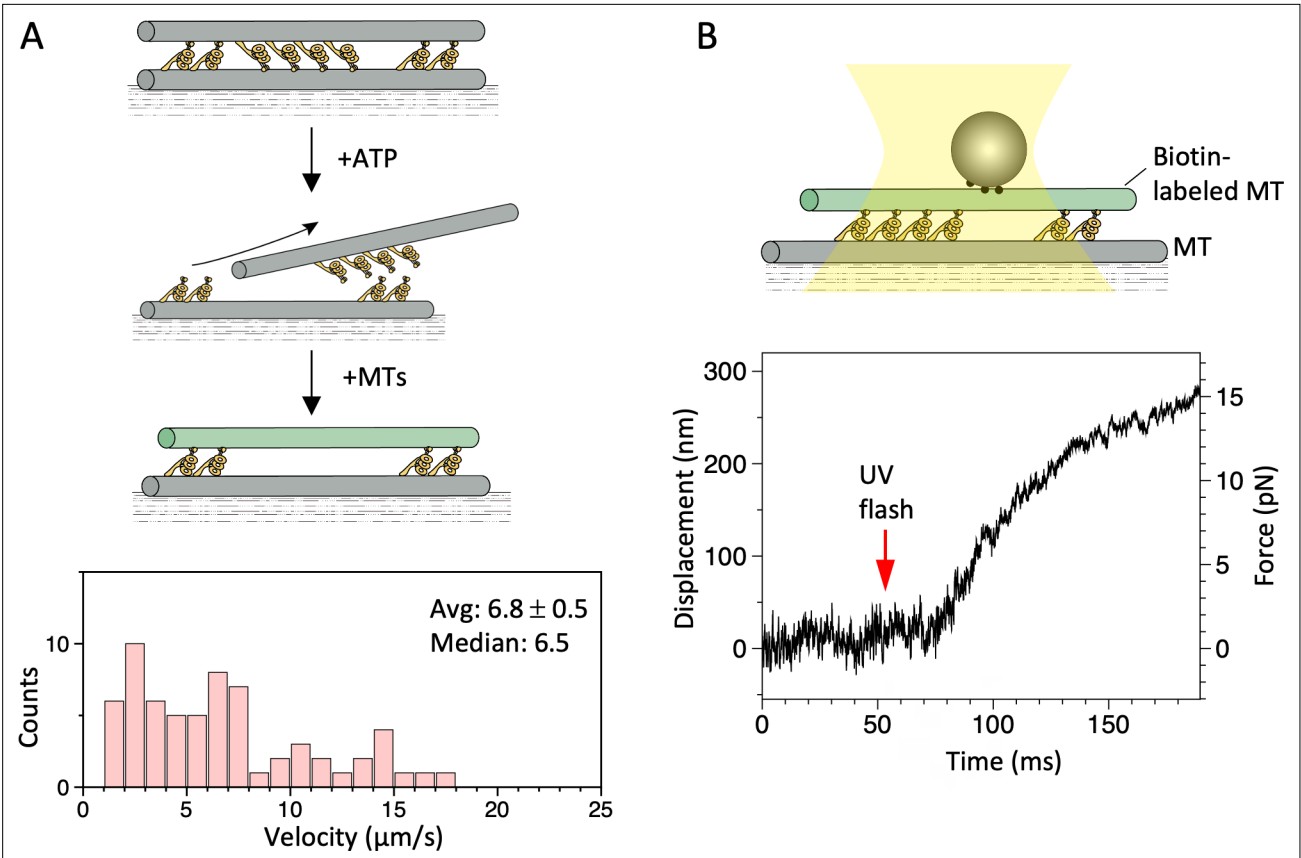

**Figure 7.** Motility of a dynein-microtubule (dynein-MT) complex in which dyneins are arranged unidirectionally. (**A**) Relative sliding of the MTs observed by fluorescence microscopy. As illustrated in the diagram, dynein-MT complexes that contain dyneins in two opposite orientations were prepared using less brightly fluorescent MTs and adsorbed to the glass. Addition of ATP disassembles the complex, leaving MTs with dyneins oriented in the same way. New MTs (more brightly fluorescent) were added in the absence of ATP to make the complexes. Average (mean ± SEM) and median values of the velocities are indicated (n=65). (**B**) Relative sliding of the MTs measured in optical trapping assays. Trap stiffness: 0.055 pN/nm. Also see *Figure 7—source data 1*.

The online version of this article includes the following source data and figure supplement(s) for figure 7:

**Source data 1.** Numerical data for *Figure 7A*.

**Figure supplement 1.** Displacement of a microtubule (MT) over dynein-coated glass surfaces measured in optical trapping assays.

**Figure supplement 2.** Displacement of a bead attached to the dynein-microtubule (dynein-MT) complex in which dyneins are arranged unidirectionally.

whereas a dynein-MT complex without the linkers does not oscillate. However, previous work using single or a few molecules of outer-arm dynein also reported bi-directional movements over a range of several tens of nanometers (*Shingyoji et al., 1998*; *Shingyoji et al., 2015*). We thus examined whether the oscillatory movement was also observed with a MT interacting with multiple dyneins but in the absence of oppositely oriented dyneins. Two experimental designs were used: optical trapping measurements of MTs gliding over dynein-coated surfaces (*Figure 7—figure supplement 1*), and dynein-MT complexes in which all the dyneins are in the same orientation (*Figure 7A*).

For the experiments in *Figure 7—figure supplement 1*, a bead (200 nm in diameter) was attached to a MT that was bound to the dynein-coated surface, and displacement after UV photolysis of caged ATP was measured in optical trapping assays. The number of dynein molecules interacting with a MT (4.7 ± 1.9 µm [mean ± SD]) was roughly estimated to be 24 at a dynein concentration of 12.5 µg/ml (see Materials and methods). In contrast to the previous results using single or a few dyneins (*Shingyoji et al., 1998*; *Shingyoji et al., 2015*), the MTs moved smoothly in one direction (*Figure 7—figure supplement 1*).

We have also examined the motility powered by a unidirectional array of dyneins. Dynein-MT-complexes that normally contain oppositely oriented dyneins were disassembled by ATP, and new MTs

were added in the absence of ATP to make dynein-MT complexes in which all the dyneins are oriented in the same way (illustrated in *Figure 7A*). The newly added MTs were biotinylated to enable binding of avidin-coated beads in optical trapping experiments and were also more brightly labeled so that they could be identified in the dynein-MT complex. In gliding assays, the MTs in these complexes moved in one direction with an average speed of 6.8 μm/s (*Figure 7A*, *Figure 7—source data 1*), which is faster than those in the usual surface-gliding assay, as reported previously (*Aoyama and Kamiya, 2010*). Optical trapping experiments also showed at a higher resolution that the movement was unidirectional (*Figure 7B* and *Figure 7—figure supplement 2*), clearly different from the traces observed with the complex containing oppositely oriented dyneins (*Figure 5A* and *Figure 5—figure supplement 1*). Thus, we conclude that an ensemble of multiple uniformly oriented dynein molecules moves a MT unidirectionally, and that two groups of oppositely oriented dyneins are required for oscillation.

## Structures of dynein in the dynein-MT-DNA-origami complex

During oscillatory movement, the two groups of oppositely oriented dynein molecules are likely to have different conformations. Although preliminary, we have extracted dynein images from the negative-stain EM images and tested if we could detect structural differences (*Figure 2—figure supplement 3*). Previous structural analysis of axonemes revealed that the heads of dynein are shifted toward the MT minus end in the presence of ATP or ADP·vanadate, or in beating flagella (*Burgess, 1995*; *Lin and Nicastro, 2018*; *Movassagh et al., 2010*; *Sale et al., 1985*; *Ueno et al., 2014*), so that the length of the dynein molecule measured along a MT increases. Thus, we have also measured the length of dynein in our negative stain images. Although the apparent length measured in 2D images is thought to be affected by the viewing angle so that the measured values are not precise, increase of the length was detected for both orientations of dyneins in the presence of ATP ($p<0.00001$, T-test), suggesting that their structures are different from the rigor structure (*Figure 2—figure supplement 4*, *Figure 2—source data 1*). No difference was detected between the average lengths of the two oppositely oriented dyneins.

Previous EM studies reported differences in the arrangement of the heads between *Chlamydomonas* outer-arm dynein in the rigor state and in the presence of ATP or ADP·vanadate (*Goodenough and Heuser, 1982*; *Lin and Nicastro, 2018*; *Movassagh et al., 2010*). Our no-nucleotide images (*Figure 2—figure supplement 3B*) seemed to agree with the 2D projection of the previously reported cryo-electron tomography structure in the apo state (*Figure 2—figure supplement 3D*; *Movassagh et al., 2010*), in which the three heads were arranged like stacked plates. However, the head arrangement was more variable in our images with ATP (*Figure 2—figure supplement 3C*). Although the current analysis using negative-stain 2D images did not allow us to prove structural changes, comparison of the two oppositely oriented dyneins suggested differences in the arrangement of the heads in the two groups. The results show that our dynein-MT-DNA-origami complex is a useful system for future structural analysis of dynein during oscillation.

## Bending motions of the dynein-MT-DNA-origami complex

In cilia and flagella, which are free to move in water, relative movements of the neighboring doublet MTs lead to bending motions. On the other hand, the dynein-MT-DNA-origami complexes in our optical trapping experiments were fixed to the glass surface along the whole length. In order to investigate whether the dynein-MT-DNA-origami complex has an ability to produce bending motions, we searched for a complex which is held at one point. Whereas a majority of the complexes seemed to be fixed to the glass along their entire length, there were some complexes attached only partially either to a bead or to the glass. Some examples are shown in *Figure 8*, *Figure 8—figure supplement 1*, *Videos 2 and 3*. Although our fluorescence microscopy setup did not allow us to detect high-frequency oscillations, repetitive bending motions were indeed observed. The plot in *Figure 8B* shows that the complex can bend in both directions compared to the position before ATP release. Thus, it is likely that the bending motions in the opposite directions are powered by oppositely oriented dyneins, not just the cycles of force production and detachment. Sometimes a part of the MT bundle separated during bending motions as in *Figure 8—figure supplement 1A*, probably because this region did not have DNA origami attachment or because the DNA origami was pulled away during bending. Using these regions, relative sliding distances of the MTs were estimated to be approximately 130 and 370

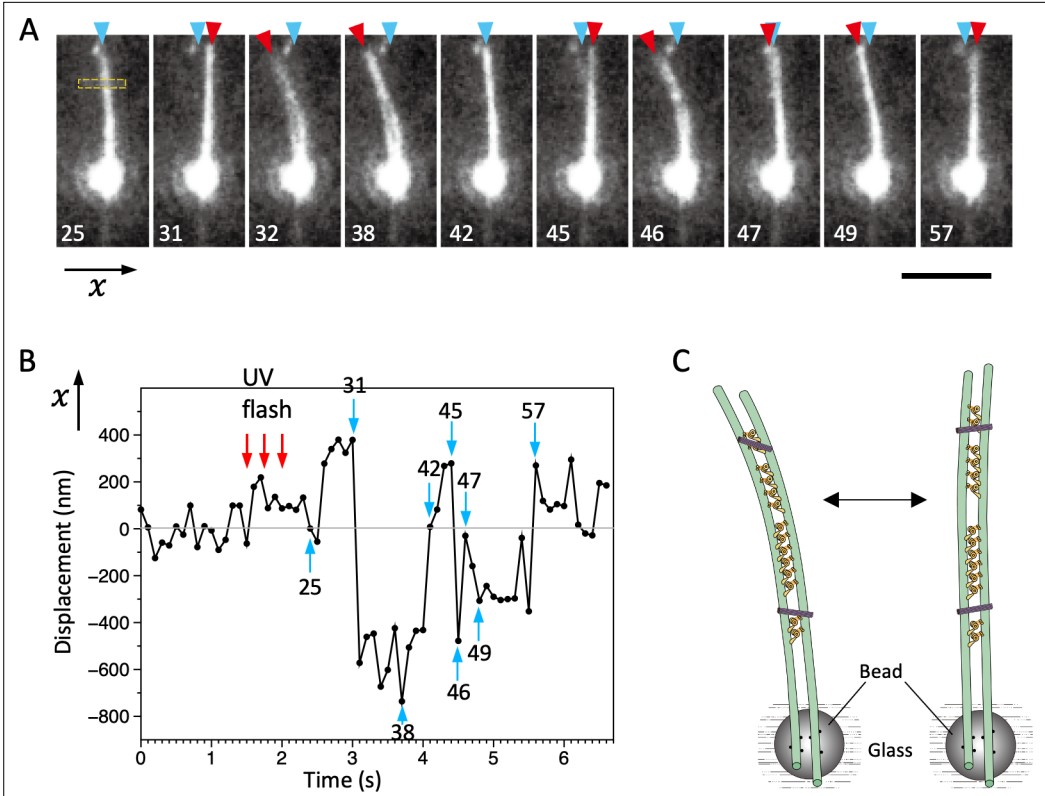

**Figure 8.** Bending motions of the dynein-microtubule-DNA-origami (dynein-MT-DNA-origami) complex.
(**A**) Snapshots during movement of the dynein-MT-DNA-origami complex shown in *Video 2*. The number indicated in each frame corresponds to the frame number in the movie (recorded at 10 frames/s). The UV was flashed at frames #16, 18, and 21, each for 50 ms. The complex is attached to a bead while one end (red arrowheads) is free, which moves with respect to the position before the UV flash (cyan arrowhead) as the complex bends repeatedly. Bar: 5 µm. (**B**) Displacement of the complex during bending motion. The plot shows the lateral (**x**) positions of the complex observed in the boxed region in A. The average position before the UV flash (frames #5–15) was taken as 0 on the vertical axis. Note that the displacement is to both the plus and minus directions. (**C**) Diagram illustrating a model explaining the movement.

The online version of this article includes the following figure supplement(s) for figure 8:

**Figure supplement 1.** An example of bending motions of the dynein-microtubule-DNA-origami (dynein-MT-DNA-origami) complex.

---

nm for the frames #38 of *Figure 8A* and #13 of *Figure 8—figure supplement 1A*, respectively. Sliding distances longer than the expected maximum value (~190 nm; see *Figure 4—figure supplement 1*) are thought to be caused by rearrangement of the DNA origami during MT sliding. Future studies at a higher resolution may reveal bending motions at higher frequencies without separation of MTs. Nevertheless, the bending motions observed here indicate that a system composed of MTs, oppositely oriented dyneins, and inter-MT crosslinkers has the ability to bend repetitively.

## Discussion

Repetitive bending of cilia and flagella is produced by collective actions of many dynein molecules that are aligned in a specific manner and communicate with each other. Another important feature of cilia and flagella is that the dyneins on the opposite sides of the axoneme produce force to bend the axoneme in opposite directions (*Figure 1A*). The dynein-MT system used here mimics cilia and flagella in that it contains two groups of oppositely oriented dynein molecules, that the neighboring dyneins are arranged with 24 nm periodicities between a pair of MTs, and that the MTs are interconnected with passive linkers. Previous work using frayed *Chlamydomonas* flagella axonemes showed that a pair of doublet MTs can display cyclic association, buckling, and dissociation (*Aoyama and Kamiya, 2005*;

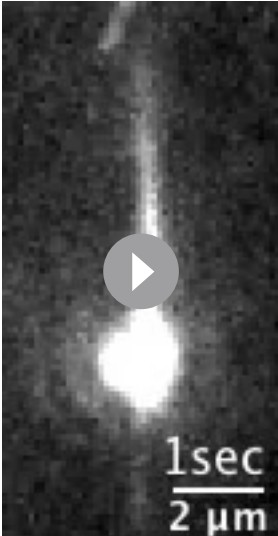

**Video 2.** Bending motions of the dynein-microtubule-DNA-origami (dynein-MT-DNA-origami) complex. Movie of the dynein-MT-DNA-origami shown in Figure 8. The complex is fixed to beads while one of the ends is free in solution. After UV photolysis of caged ATP (frames #16, 19, 21), the complex bends repeatedly. Recorded at 100 ms/frame.

https://elifesciences.org/articles/76357/figures#video2

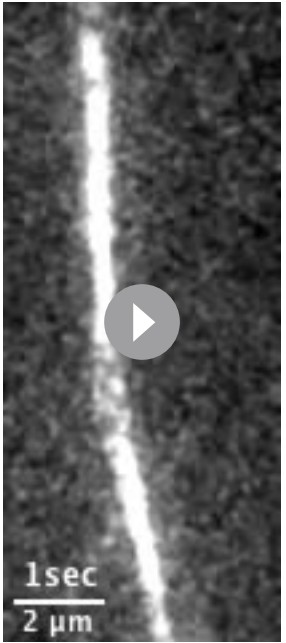

**Video 3.** Bending motions of the dynein-microtubule-DNA-origami (dynein-MT-DNA-origami) complex. Movie of the dynein-MT-DNA-origami shown in Figure 8—figure supplement 1. Movement of the dynein-MT-DNA-origami complex was recorded under the same conditions as in *Video 2* but without beads. One end of the complex is attached to the glass surface, while the MTs in the upper part of the image slide relative to each other, which causes bending and separation of the MTs in the middle part. UV flash at frames #5, 15, and 25. Recorded at 100 ms/frame.

https://elifesciences.org/articles/76357/figures#video3

*Brokaw, 2009*). In their system, the doublet MTs were bundled at the proximal end and the inter-doublet linkers were proteolyzed. Our dynein-MT complex differs from theirs in that it contains inter-MT linkers and two groups of dyneins that produce opposing forces between a pair of MTs.

Oscillatory movements of MTs were also observed by *Shingyoji et al., 1998*; *Shingyoji et al., 2015*, using single or a few molecules of sea urchin axonemal dynein. In contrast, our work shows that multiple molecules of *Chlamydomonas* outer-arm dynein move a MT unidirectionally. We think the discrepancy is caused by the number and the arrangement of molecules: our system has multiple dynein molecules (~35 on average in the case of the complex with oppositely oriented dyneins) aligned regularly on a MT, which would work cooperatively as in cilia and flagella. An effect of the number of molecules on a motor's directionality was also reported for kinesin-5 Cin8: a single Cin8 moved to the MT minus end, while a team of Cin8 showed plus-end-directed motility (*Roostalu et al., 2011*). Our results show that an ensemble of dynein molecules aligned on a MT produce force in one direction, but participation of oppositely oriented dyneins is required for oscillatory movements.

The axoneme is a complex structure containing various components that may modulate the dynein activity, which makes it difficult to understand the basic mechanism of oscillation. For example, radial spokes and central apparatus are thought to be important for beating of flagella, but mutant flagella lacking radial spokes or the central pair can beat under certain conditions (*Frey et al., 1997*; *Yagi and Kamiya, 2000*), indicating that they are not absolutely needed for oscillatory beating. The circumferential layout of nine doublet MTs and/or the cooperative activities of outer-arm and several species of inner-arm dyneins may also be important for oscillation, but it has not been known whether they are essential for oscillation. Some modeling studies predicted that a system composed of dynein, MTs, and passive components can produce oscillation due to mechanical instability (*Bayly and Dutcher, 2016*), but direct evidence has not been obtained because of the lack of a reconstituted system with minimal components. Here, use of DNA origami enabled us to construct a simple system composed of two MTs with clusters of oppositely oriented dyneins and passive linkers. Oscillation of this system

shows that the regulatory components including the 9 + 2 structure with radial spokes and the central apparatus, nexin-dynein regulatory complexes, and inner-arm dyneins are not essential for generation of oscillation per se.

Since our dynein-MT-DNA-origami complex contained dyneins that produce forces in opposite directions, the most likely explanation for the oscillatory movements would be that each of the two dynein orientation-groups is responsible for just one of the two directions of movement, one orientation powering forward movements and the other orientation driving backward movements (*Figure 4*). We cannot completely rule out the possibility that at least some of the backward motions are caused by the trap force after detachment of dyneins; although steps were observed in the backward motions, force-induced backward stepping was reported with cytoplasmic dynein (*Gennerich et al., 2007*). Nevertheless, we think that it is more likely that the oscillatory movements are produced by alternating actions of two groups of dyneins that produce force in opposite directions, because (1) the dynein-MT complexes with unidirectionally aligned dyneins did not show oscillation and (2) there exist two groups of dyneins that produce opposing force in more than 80% of the complexes, and if all the steps observed during backward motion were force-induced backward stepping, that would mean the dyneins in only one of the two groups are responsible for both forward and backward movement while the dyneins in the other group do nothing, which we think is unlikely.

The DNA linkers limit the range of relative sliding, and thus may function as the trigger to change the direction of movement. The average frequency of the observed oscillation (~33 Hz) was close to the beat frequencies of the two flagella of a detergent-extracted *Chlamydomonas* cell (30 and 45 Hz) (*Kamiya and Hasegawa, 1987*). In addition, the dynein-MT-DNA-origami complex showed repetitive bending motions when one end is free in solution. Although regulatory components including the radial spokes and central apparatus are probably needed for more coordinated and long-lasting oscillation, our results show that a minimal system composed of two groups of oppositely oriented dyneins, MTs, and inter-MT crosslinkers has an intrinsic ability to oscillate and bend cyclically.

The dynein-MT-DNA-origami complex developed here contains multiple dynein molecules aligned between the MTs with a 24 nm periodicity as in axonemes, allowing the neighboring dyneins to cooperate, and the oppositely oriented dyneins to work against each other. This simple system is thus useful for future research on the mechanism of cilia/flagella motility. It can also be used to study the high-resolution structures of dynein during oscillatory movement. Future cryo-EM studies analyzing the 3D structures of activated and regulated dynein molecules would be of great interest.

## Materials and methods
### Dynein, kinesin, and microtubules
A wild-type strain (137c-) of *Chlamydomonas reinhardtii* was used. Flagella were isolated using dibucaine and demembranated as described previously (*Yagi et al., 2009*). For extraction of dynein, axonemes were incubated in 0.6 M KCl in HMDE solution (30 mM HEPES, 5 mM $MgSO_4$, 1 mM dithiothreitol, 1 mM EGTA, pH = 7.4) for 30 min at 4°C. For some preparations, axonemes were first incubated in HMDEK (HMDE with 50 mM $CH_3COOK$) containing 0.1 mM ADP for 1 min, centrifuged, and then incubated in 0.6 M KCl in HMDE for 20 min two times. After centrifugation, the excess salt was removed from the supernatant fraction either by overnight dialysis or by Amicon Ultra-4 100 K centrifugal filter using HMDEK. The resulting dynein suspension was mixed with 23% sucrose, aliquoted, and stored in liquid nitrogen.

The E237A mutant of human ubiquitous kinesin, which is ATPase-defective with the neck-linker in a docked conformation (*Rice et al., 1999*), was used. SNAPf protein was fused to the C-terminus of cysteine-light, 336-residue monomer kinesin E237A. All constructs were verified by DNA sequencing. Monomeric kinesins were expressed and purified as previously described (*Miyazono et al., 2010*), except that kinesins were further purified with HiTrap-SP (GE) after His-tag purification (Ni-NTA Agarose, Qiagen).

Tubulin was purified from porcine brain tissue (*Castoldi and Popov, 2003*). Rhodamine-labeled and biotinylated tubulin was prepared using tetramethylrhodamine (C-1171, Molecular Probes) and Sulfo-NHS-LC-LC-Biotin (Thermo Scientific), respectively. For fluorescently labeled MTs, rhodamine-labeled tubulin and unlabeled tubulin were mixed in the ratio 1:9. For MTs used to bind beads in the optical trapping experiments, rhodamine-labeled tubulin, biotinylated tubulin, and unlabeled tubulin

were mixed in the ratio 1:2:7. MTs were polymerized in a polymerizing solution (80 mM PIPES (pH 6.8), 1 mM EGTA, 5 mM MgSO$_4$, 1 mM DTT, 0.5 mM GTP, and 5 or 10% DMSO) for 30 min, stabilized with taxol and stored in liquid nitrogen.

### DNA origami

Single-stranded P8064 DNA purchased from tilibit nanosystems GmbH (Garching, Germany) was used as the scaffold for DNA origami. Unmodified staple strands were purchased from Sigma-Genosys as Oligonucleotide Purification Cartridge (OPC) grade. Amino-modified staples, which have three amino-modified C6dT oligonucleotides near the 5' end, were purchased from IDT as Dual-HPLC-purified, or from Japan Bio Service (Tsukuba, Japan) as HPLC-purified. The SNAP-ligand was covalently attached to the amino-modified staples by mixing the staples and BG-GLA-NHS (NEB, dissolved in DMSO) as previously described (*Derr et al., 2012*; *Masubuchi et al., 2018*). The label efficiency of the SNAP-ligand was estimated to be 95–98% by a gel-shift assay using purified SNAP-tag protein. The modified staples are described in *Supplementary file 1*.

The rod type DNA origami nanostructure composed of 30 helices was designed using the honeycomb-lattice version (10.5 bp/turn) of the caDNAno2 software, and folded in 1 × Rod buffer (5 mM Tris boric acid, pH 7.6, 20 mM Mg(OAc)$_2$, 5 mM Na(OAc), and 1 mM EDTA). Typically, 40 nM single-stranded P8064 DNA (tilibit) and 240 nM of each staple strand (sixfold excess) were mixed in 1 × Rod buffer, denatured at 85°C for 5 min and annealed at 47.5°C for 4 hr in a PCR machine (Takara-Bio). The folded DNA origami was then agarose-gel-purified (*Douglas et al., 2009*) and concentrated by PEG-precipitation (*Stahl et al., 2014*). The Rod concentration was estimated by absorbance at 280 nm using NanoDrop (Thermo).

The maximum length of the linker was estimated assuming the following: the unit length of ssDNA: 0.63 nm, the unit length of amino acids: 0.34 nm, the unit length of a carbon chain: 0.13 nm, and the size of the SNAPf part: 2 nm. Using these values, the maximum length of the linker including 30 nucleotides of poly-thymidine, C6dT oligonucleotide, SNAPf, and an amino acid linker between kinesin and SNAPf, was calculated to be approximately 29 nm.

### Dynein-MT-DNA-origami complex

About 15–20 nM DNA rods were incubated with kinesin E237A at a molar ratio of 1:8 in HEM buffer (20 mM HEPES, 1 mM EGTA, 10 mM MgSO$_4$, pH = 7.8) for 30 min at room temperature. Meanwhile, the dynein-MT complex was prepared by incubating 10–20 µg/ml dynein preparation with ~40 µg/ml MTs in HEM for 8–10 min at room temperature. Dynein-MT and DNA-kinesin were then mixed so that the final concentrations of dynein, MTs, DNA, and kinesin were 6.25 or 12.5 µg/ml, 25 µg/ml, 2.5 nM, and 20 nM, respectively, and incubated for 10 min at room temperature. For the dynein-MT complex without DNA-kinesin, HEM was added instead of DNA-kinesin.

### Preparation of the dynein-MT-DNA complex with a unidirectional array of dynein

For preparation of the dynein-MT complex that has dyneins oriented in the same way (*Figure 7*), dynein-MT complexes were prepared using less-brightly fluorescent MTs (rhodamine tubulin: unlabeled tubulin = 3:97) at a final concentration of dynein and MTs of 30 and 15 µg/ml, respectively, and adsorbed to the glass surface of a chamber. The complexes, which are thought to contain oppositely oriented dyneins, were then separated by addition of 0.5 mM ATP in HEM. After perfusion of HEM containing 1 mM ADP, brighter MTs (rhodamine tubulin:biotinylated tubulin:unlabeled tubulin = 1:2:7) were added to make dynein-MT complexes that contain dyneins in a single orientation. Biotinylated tubulin was used for binding to avidin-coated beads in optical trapping experiments.

### EM and image analysis

The sample was applied onto a carbon-coated grid and stained with 1–2% uranyl acetate. For observing the complex in the presence of ATP, ADP (final: 1 mM) was first added to the sample on a grid (*Yagi, 2000*). After 1 min, ATP (final: 0.1 mM) was added and the sample was immediately stained with uranyl acetate. The samples were observed using an FEI Tecnai F-20 electron microscope equipped with a Gatan Orius 831 CCD camera. The images were adjusted for contrast and Gaussian-filtered in Adobe Photoshop to reduce noise.

For the analysis of dynein shapes shown in *Figure 2—figure supplement 3*, a set of high magnification images and a low magnification image were taken from the same dynein-MT-DNA-origami complex. The high magnification images show the dynein structures at a higher resolution and comparison with the low magnification image tells where each dynein is located in the complex.

The electron microscopic images were analyzed using the Eos software (*Yasunaga and Wakabayashi, 1996*). Images of dynein-MT-DNA-origami complexes that have dynein molecules bound in two opposite orientations were selected. After correction for the contrast transfer function, image segments each containing a dynein molecule and parts of two MTs were extracted (*Figure 2—figure supplement 3A*). Images of dynein molecules whose tails are bound to the same MT were grouped, so that there are two groups of dynein images observed between a pair of MTs.

Image segments that belong to the same group were aligned and averaged in the following process. To lower the effect of the MTs and neighboring dyneins during alignment, the contrast of the peripheral region in each segment was reduced by 10%. Each segment was pasted into a slightly larger box filled with the mean density of the peripheral region. For each group, we selected one of the segments as the original reference and aligned the other images by calculating their similarity using a correlation function, allowing translation and rotation with 9–0.072° steps. The images in the same group were averaged using the rotation angles and translation values that showed the maximum correlation. We then used these average images as the references for the next cycle of alignment. The alignment cycle was repeated until the average images did not change. For comparison with these average images, 2D projections of the EMDB maps (EMD_1696 and EMD_1697) (*Movassagh et al., 2010*) were calculated using mrcImageRotation3D/mrcImageProjection on Eos (*Yasunaga and Wakabayashi, 1996*).

The length of the dynein molecule along the MT axis was measured in each average image. The length of the dynein image projected onto the MT axis was defined as the dynein length (*Figure 2—figure supplement 4*, inset). Distributions of the lengths in the presence and absence of ATP were compared statistically (Welch's T-test; *Figure 2—source data 1*).

## In vitro motility assays

Glass chambers were made using two pieces of cover glass (24 mm × 32 mm [No. 24321, Muto pure chemicals, Japan] and 18 mm × 18 mm [No. 0101030, Marienfeld]). The glass was cleaned with KOH, and then rinsed with $H_2O$ and ethanol for MT-gliding assays (*Inoue and Shingyoji, 2007*). For optical trapping experiments of the dynein-MT complex, uncleaned glass was used because more beads adhered non-specifically to the cleaned glass. The size of the chamber was ~18 mm × 5 mm.

MT-gliding assays over dynein-coated glass surfaces (*Figure 3A*) were performed by introducing 100–125 µg/ml dynein in a chamber for 2 min, followed by 1 mM ADP in HEM for 1 min, and then 5 µg/ml MT in the assay buffer (1 mM caged ATP [Dojin], 0.1 mM taxol, 20 mM glucose, 0.5% [v/v]; β-mercaptoethanol, 20 mg/ml catalase, 100 µg/ml glucose oxidase, and 1 unit/ml hexokinase in HEM). Movement of MTs after UV photolysis of caged ATP was observed using a fluorescence microscope (81 X, Olympus, Japan) equipped with a 100× lens, a mercury lamp (USH1030L), and filters (535 BP/30 and 330 WB/80). Caged ATP was photolyzed by switching the filters. Images were recorded using a CMOS camera (ORCA-Flash 2.8, Hamamatsu Photonics, Japan).

For motility assays of the dynein-MT complex with and without DNA (*Figure 3B*, *Video 1*, and *Figure 4—video 1*), the complex preformed in solution was perfused into the chamber. After applying the ADP solution and assay buffer (without MTs), the MT movement after caged ATP photolysis was observed as above. The velocities were calculated from the videos recorded at the frame rates of 100–400ms.

## Optical trapping

Optical trapping experiments were performed as previously described using a custom-made microscope system (*Kinoshita et al., 2018*). After perfusion of the dynein MT-complex with and without DNA-kinesin, 1 mM ADP in the HEM buffer, and then the assay buffer (1 mM caged ATP [Dojin], 0.1 mM taxol, 20 mM glucose, 0.5% [v/v] β-mercaptoethanol, 20 mg/ml catalase, 100 µg/ml glucose oxidase, and 1 unit/ml hexokinase in HEM) with streptavidin-coated beads and 0.2 mg/ml casein were introduced into the glass chamber.

An avidin-coated bead was trapped and then brought in contact with a biotinylated MT in the complex. Caged ATP was photolyzed by a UV laser pulse. Displacement of the bead was measured at a sampling rate of 20 kHz with a quadrant photodiode connected to a MacLab system (AD Instruments). For the experiments with dynein-MT-DNA-origami complexes, 200 nm beads were used (trap stiffness: 0.015–0.055 pN/nm). For measuring the maximum force produced by the dynein array in the dynein-MT complex (*Figure 3—figure supplement 1*), 500 nm beads were used (trap stiffness: 0.175–0.35 pN/nm), because the bead-attached MT often slid out of the optical trap and the dynein-MT complex disassembled when 200 nm beads were used.

For the optical trapping experiments of MTs gliding over dynein-coated surfaces (*Figure 7—figure supplement 1*), 10 µl of 125 µg/ml dynein was applied to the glass chamber as in usual MT-gliding assays. Assuming that 10% of the perfused dynein molecules are adsorbed to the glass (*Sakakibara et al., 1999*), the density of the dynein molecules was roughly estimated to be 209 molecules/µm². Thus, if dynein molecules directly below a MT that has a diameter of 25 nm can interact with the MT, the number of dynein molecules that interact with the MT is calculated to be 5.2 molecules/µm.

Stepping events were analyzed by a step-finding algorithm by *Kerssemakers et al., 2006*. To statistically optimize the step finding procedure, they defined a parameter $S$, which is the ratio of the $\chi^2$ of the counter fit to $\chi^2$ of the best fit. The $S$ changes with number of steps, and the highest peak S gives the best fit. For the analysis shown in *Figure 6F*, only those data that had a clear peak in $S$ were used. The detected step sizes were calibrated with attenuation factors to account for the compliance of the bead-MT linkage and that of dynein (*Kinoshita et al., 2018*). Fitting with Gaussian functions was performed by graphic software (Datagraph, Visual Data Tools, Inc).

## Observation of bending motions

Avidin-coated beads (200 nm in diameter) were perfused into a glass chamber. After washing with HEM buffer, dynein-MT-DNA-origami complexes prepared as above using 6.25 µg/ml (final) dynein and rhodamine-labeled MTs were introduced into a chamber and incubated for 5 min. The assay buffer containing 1 mM caged ATP, 0.5 mM ADP, and 0.3 mg/ml casein was then perfused and the sample was observed under a fluorescence microscope. Although the majority of complexes firmly attached to the glass and did not show movement, some complexes appeared to be attached only by a part of their length while the other ends were free and showed movement. For some experiments, avidin-coated beads were first introduced to the chamber, and dynein-MT-DNA-origami complexes made with rhodamine and biotin-labeled MTs were bound to the beads.

Positions of the MTs were detected by fitting Gaussian functions to the fluorescence intensity profile using ImageJ. The maximum values of relative sliding of the MTs in the complex were estimated using the parts where two MTs of the complex separated (e.g. shown by orange arrowheads in *Figure 8—figure supplement 1A*). The length along each MT was measured and the difference between the lengths of the two MTs was taken as the sliding distance.

## Acknowledgements

We are grateful to Toshiki Yagi, Ken-ichi Wakabayashi and Ritsu Kamiya for the gift of Chlamydomonas strains and continuous advice on dynein preparation, Chikako Shingyoji and Izumi Nakano for guidance on motility assays, Akira Nagasaki for advice on fluorescence microscopy, Yoshie Harada for support, and Hironori Ueno for initial experiments with the dynein-MT system. This work was supported by JSPS KAKENHI Grant Numbers 24115522, 16K07332, 17H05898 for KH, 15H00798, 19H03197 for HT, 16H04773, 19H03189 for HH, JPMJCR1865 for TY.

## Additional information

### Funding

| Funder | Grant reference number | Author |
| --- | --- | --- |
| Japan Society for the Promotion of Science | 24115522 | Keiko Hirose |

| Funder | Grant reference number | Author |
|---|---|---|
| Japan Society for the Promotion of Science | 16K07332 | Keiko Hirose |
| Japan Society for the Promotion of Science | 17H05898 | Keiko Hirose |
| Japan Society for the Promotion of Science | 15H00798 | Hisashi Tadakuma |
| Japan Society for the Promotion of Science | 19H03197 | Hisashi Tadakuma |
| Japan Society for the Promotion of Science | 16H04773 | Hideo Higuchi |
| Japan Society for the Promotion of Science | 19H03189 | Hideo Higuchi |
| Japan Society for the Promotion of Science | JPMJCR1865 | Takuo Yasunaga |

The funders had no role in study design, data collection and interpretation, or the decision to submit the work for publication.

### Author contributions

Shimaa A Abdellatef, Formal analysis, Investigation, Writing – original draft; Hisashi Tadakuma, Funding acquisition, Investigation, Methodology, Writing – original draft; Kangmin Yan, Kodai Fukumoto, Rofia Boudria, Investigation; Takashi Fujiwara, Hiroko Takazaki, Formal analysis; Yuichi Kondo, Formal analysis, Investigation; Takuo Yasunaga, Formal analysis, Funding acquisition; Hideo Higuchi, Formal analysis, Funding acquisition, Investigation, Supervision; Keiko Hirose, Conceptualization, Formal analysis, Funding acquisition, Investigation, Project administration, Writing – original draft, Writing – review and editing, Supervision

### Author ORCIDs

Hideo Higuchi http://orcid.org/0000-0002-6519-7348
Keiko Hirose http://orcid.org/0000-0003-4587-8346

### Decision letter and Author response

Decision letter https://doi.org/10.7554/eLife.76357.sa1
Author response https://doi.org/10.7554/eLife.76357.sa2

# Additional files

### Supplementary files

• Supplementary file 1. DNA oligonucleotide sequences for the DNA origami rod. The asterisk indicates the C6dT modified oligonucleotide. Small letter "tttt" indicates the four thymine (4T) linker added to prevent the aggregation of DNA rods.

• Transparent reporting form

### Data availability

Figure 2 - source data 1, Figure 3 - source data 1, Figure 4 - source data 1, Figure 4 - source data 2, Figure 6 - source data 1, and Figure 7 - source data 1 contain the numerical data used to generate the figures and statistical analysis. Supplementary file 1 contains the DNA sequence for the DNA origami.

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
