## [Editor Report]

The authors describe the reconstitution of axonemal bending using polymerized microtubules, purified outer-arm dyneins, and synthesized DNA origami to cross-link two microtubules. The work is of interest to the field as it shows that bidirectional sliding and bending of microtubules can be generated by a minimal set of elements.

---

## [Decision Letter]

**Decision letter after peer review:**

Thank you for submitting your article "Oscillatory movement of a dynein-microtubule complex crosslinked with DNA origami" for consideration by *eLife*. Your article has been reviewed by 3 peer reviewers, and the evaluation has been overseen by a Reviewing Editor and Anna Akhmanova as the Senior Editor. The following individual involved in review of your submission has agreed to reveal their identity: Robert Cross (Reviewer #3).

Essential revisions:

1) Address Reviewer #2's concerns about the EM characterization of your system.

2) Address Reviewer #3's concerns. An implication of this point is that the DNA-based cross-bridges may not have cross-linked the two microtubules.

3) Please clarify if each measured motion is sawtooth-like or oscillatory and if it can be explained by your model.

I include all the reviewer's comments below to help with your revisions.

*Reviewer #1 (Recommendations for the authors):*

The described findings should be of interest for the cytoskeletal cell and biophysics communities. I therefore recommend publication, subject to the following changes:

1. Abstract: The description that "…axonemal dynein molecules move in an oscillatory manner along a microtubule…" is not correct. All dynein molecules perform unidirectional power strokes along axonemal MTs (they don't oscillate along MTs). The oscillatory beating of axonemes results from the coordinated, out-of-phase activation/deactivation of dynein molecules at opposing sides of the axoneme. I recommend correcting the first sentence and the last sentence on page 2.

2. Introduction: References are missing at the end of "Beating of cilia and flagella is powered…", "…the inner-arm dyneins are also regularly placed in specific positions", and "…produced by some dyneins to the other parts of the axoneme".

3. Results, page 6: The authors state that the demonstrated activity shows oscillatory motion. The depicted displacements in Figure 5 and Suppl. Figure 5 look mostly like unidirectional force generation followed by detachment (more than 90% of the depicted events show this behavior). This sawtooth-like pattern is even expected as the force generation of the dynein motors must eventually cease as the MTs are cross-linked. Sections where backward motion is observed could be force-induced backward stepping. The authors should discuss the difference between the observed sawtooth-like displacements and the true oscillatory beating of cilia and flagella.

4. Video 2: I am not sure what one is supposed to see in Video 2.

5. Figure 3: Please plot the absolute sliding velocities in Figure 3B and not the relative velocities.

6. Figures 5B and Suppl. Figure 5A: Why can 180-200 nm unidirectional displacements be generated if both MTs are cross-linked by DNA origami? What explains these large displacements considering that the following sawtooth-like displacements are only 50 nm in amplitude?

7. Figure 7: It looks like the trace in Figure 7B shows also sawtooth-like displacements initially after the photolysis of caged ATP, which can also be seen in Supplemental Figure 7C. As the amplitude of these displacements is on a similar length scale (~50 nm) as reported by Shingyoji et al., (Nature 1998), I would say that these experiments largely agree with the results of Shingyoji and co-workers.

*Reviewer #2 (Recommendations for the authors):*

The authors could discuss how the force generating dyneins are localized in cilia. In Figure 8c, dynein generates force at the straight parts, not at the bending point. This is close to the model based on the sliding disintegration (Hayashi and Shingyoji, 2008) and different from the one proposed based on cyo-ET (Lin and Nicastro), where asymmetrical dynein activation occurs at the bending points.

*Reviewer #3 (Recommendations for the authors):*

1. Please explain in the Abstract that the paired microtubules are arranged in parallel, otherwise it is hard to understand that 'opposite orientation' (of the dynein sets) refers to their originating on different microtubules.

2. I don't understand why restraining force due to the DNA crosslinker triggers oscillatory switching between the two sets of dyneins, whereas restraining force due to the optical trap does not. I may be missing something, please comment?

3. I don't understand why in Figure 5B, the initial response to the UV flash is 170 nm of unidirectional sliding, after which much smaller amplitude oscillations ensue. Were the DNA linkers not initially engaged? Can the dynein drag the origami? According to Figure 4 supp1, 96 nm is the max possible sliding distance. Again I may be missing something. There are similar large amplitude sliding events in some of the traces in Figure 5 supp 1.

4. l.186 How are oscillation events defined?

5. Does the steppiness of the traces imply that relatively few dyneins (perhaps sometimes only one?) are engaged at any one time? Or are multiple dyneins synchronising? In Fig7 Supp2 (no origami, larger forces) are the traces steppy?

6. Concerning the possible difference in the velocity distribution for backwards sliding versus forwards sliding (relative to the trap centre) (Figure 6C), the authors consider the possibility that the trapping potential may be hindering the one and helping the other, and raise the possibility that some backwards motions are slips. Can these events not be recognised according to their special rapidity? If so, are there any 'forwards' detachment events?

---

## [Author Response]

Essential revisions:1) Address Reviewer #2's concerns about the EM characterization of your system.

In response, we have added some more analysis and revised the manuscript as we wrote as response to Reviewer #2’s comments.

2) Address Reviewer #3's concerns. An implication of this point is that the DNA-based cross-bridges may not have cross-linked the two microtubules.

Please see our response to Reviewer #3’s *‘*Recommendations for the authors #3’.

3) Please clarify if each measured motion is sawtooth-like or oscillatory and if it can be explained by your model.

Please see our response to Reviewer #1’s *‘*Recommendations for the authors #3’.

Reviewer #1 (Recommendations for the authors):The described findings should be of interest for the cytoskeletal cell and biophysics communities. I therefore recommend publication, subject to the following changes:1. Abstract: The description that "…axonemal dynein molecules move in an oscillatory manner along a microtubule…" is not correct. All dynein molecules perform unidirectional power strokes along axonemal MTs (they don't oscillate along MTs). The oscillatory beating of axonemes results from the coordinated, out-of-phase activation/deactivation of dynein molecules at opposing sides of the axoneme. I recommend correcting the first sentence and the last sentence on page 2.

Thank you for this comment. We did not intend to say that the dynein molecules actively move back and forth. The axoneme bends in one direction when dynein molecules in a part of the axoneme produce force and move towards the minus end. But then, these dyneins stop moving and are pulled back as the axoneme bends in the other direction. Thus, a dynein molecule moves back and forth, although the backwards movements are probably passive.

To make our intention more clearly understood, we have added a drawing as new Figure 1A, and modified the sentences the reviewer suggested.

First sentence of Abstract: “Bending of cilia and flagella occurs when axonemal dynein molecules on one side of the axoneme produce force and move toward the microtubule (MT) minus end. These dyneins are then pulled back when the axoneme bends in the other direction, meaning oscillatory back and forth movement of dynein during repetitive bending of cilia / flagella.”

Last sentence of the first paragraph of Introduction: “How a minus-end-directed motor dynein can move back and forth along a MT is unknown.”

2. Introduction: References are missing at the end of "Beating of cilia and flagella is powered…", "…the inner-arm dyneins are also regularly placed in specific positions", and "…produced by some dyneins to the other parts of the axoneme".

We have added references.

3. Results, page 6: The authors state that the demonstrated activity shows oscillatory motion. The depicted displacements in Figure 5 and Suppl. Figure 5 look mostly like unidirectional force generation followed by detachment (more than 90% of the depicted events show this behavior). This sawtooth-like pattern is even expected as the force generation of the dynein motors must eventually cease as the MTs are cross-linked. Sections where backward motion is observed could be force-induced backward stepping. The authors should discuss the difference between the observed sawtooth-like displacements and the true oscillatory beating of cilia and flagella.

According to our understanding, a ‘sawtooth pattern’ is an upward movement followed by a sharp drop caused by detachment, as seen from the first 60 ms to 170 ms of the trace in Figure 5. This reviewer suggests that ‘more than 90% of the depicted events show this behavior’ (unidirectional force generation followed by detachment). We do not agree with this comment, because, for example, 3 out or 5 backwards movements indicated by pink arrows in Figure 5 show steps. Also, the average velocity of the backwards movements (6.4 µm/s; Figure 6D) is much slower than the velocity after detachment (>600 µm/s during detachment that happens after 160 ms in Figure 5). We have added another example of steps in Figure 5 – suppl 1, where backwards movements during oscillation (indicated by pink arrows) show steps. We have also added a histogram to compare the time required for forwards and backwards movements in new Figure 6. The average time required for backwards movement (8.3 ms) was slightly shorter than that for forwards movement (10.1 ms) but much longer than the time required for simple detachment from a MT (<1 ms).

We realize that we cannot completely rule out the possibility that the observed steps during the backwards movement are ‘force-induced backward stepping’. Nevertheless, we think that the most likely explanation for the oscillatory movements would be that each of the two dynein orientation-groups is responsible for just one of the two directions of movement, one orientation powering forwards movements and the other orientation driving backwards movements, because 1) the dynein-MT complex with unidirectionally aligned dynein did not show oscillation; and 2) there exist two groups of dyneins that produce opposing force in more than 80 % of the complexes, and if the steps observed during backwards motion are force-induced backwards stepping, that would mean the dyneins in only one of the two groups are responsible for both forwards and backwards movement while the dyneins in the other group do nothing, and we cannot think of any reason for that.

In the revised manuscript, we have added discussion about the possibility that the observed backwards movements might be ‘force-induced backward stepping’, and the reasons why we think they are caused by oppositely oriented dyneins:

“We cannot completely rule out the possibility that at least some of the backwards motions are caused by the trap force after detachment of dyneins; although steps were observed in the backwards motions, force-induced backwards stepping was reported with cytoplasmic dynein (Gennerich *et al.*, 2007). Nevertheless, we think that it is more likely that the oscillatory movements are produced by alternating actions of two groups of dyneins that produce force in opposite directions, because 1) the dynein-MT complexes with unidirectionally aligned dyneins did not show oscillation; and 2) there exist two groups of dyneins that produce opposing force in more than 80 % of the complexes, and if all the steps observed during backwards motion were force-induced backwards stepping, that would mean the dyneins in only one of the two groups are responsible for both forwards and backwards movement while the dyneins in the other group do nothing, which we think is unlikely.”

4. Video 2: I am not sure what one is supposed to see in Video 2.

Whereas the MTs slide after caged ATP photolysis in Video 1, the MTs stay still in Video 2 (Figure 4-video 1 in the revised manuscript), which means that addition of DNA origami can stop large relative sliding of the MTs and prevents disassembly of the complex. We moved this video to supplements (Figure 4-video 1), since it gives rather supporting evidence. We have also added the following phrase in line 172-173 for clarification:

“so that movement of MTs was not detected by fluorescence microscopy”.

5. Figure 3: Please plot the absolute sliding velocities in Figure 3B and not the relative velocities.

As we stated in line 135-137, ‘when the dynein-MT complexes are adsorbed to a glass surface, the MTs that are directly attached to the glass cannot move, but other MTs in the same complex are allowed to slide relative to the glass-attached MTs.’ We measured the velocity only when one MT moved while the rest of the complex stayed still. Therefore, the absolute sliding velocities are equivalent to the relative velocities at this resolution.

6. Figures 5B and Suppl. Figure 5A: Why can 180-200 nm unidirectional displacements be generated if both MTs are cross-linked by DNA origami? What explains these large displacements considering that the following sawtooth-like displacements are only 50 nm in amplitude?

A similar comment was made by Reviewer #3 (Recommendations for the authors #3). As we explained in Figure 4 —figure supplement 1A, the maximum displacement of the MT before the DNA linker becomes fully stretched depends on the initial binding angle of the DNA origami with respect to the MTs. The largest amplitude of the back-and-forth movement of a MT is expected to be ~190 nm (96 x 2 nm), but it would happen only when the binding angles of all the DNA origami rods (we estimated there are about 7 DNA rods crosslinking the complex) are equal. Therefore, the sliding distance is usually shorter than 190 nm.

However, as the MT slides, one of the origami rods that binds with the most disadvantageous tilt angle would be stretched more than the rest of the origami rods. The origami rod is bound to the MT via kinesin heads, but the binding force of one or two kinesins may not be enough for stopping the MTs sliding. These kinesins may be dragged, allowing farther sliding. The sliding would stop when enough DNA rods work together.

7. Figure 7: It looks like the trace in Figure 7B shows also sawtooth-like displacements initially after the photolysis of caged ATP, which can also be seen in Supplemental Figure 7C. As the amplitude of these displacements is on a similar length scale (~50 nm) as reported by Shingyoji et al., (Nature 1998), I would say that these experiments largely agree with the results of Shingyoji and co-workers.

The traces shown in Figure 7B and Figure 7-suppl 2C were noisy in the first 80 or 90 ms of the figure, probably because the attachment of the bead to the MT was not stiff until the bead started moving. In Author response image 1, we show these two traces starting 60~80 ms before the UV flash. The traces show similar irregular noises both before and after the UV flash. The oscillation reported by Shingyoji et al., had the frequency of ~70 Hz, corresponding to ~14 ms for one up-and-down cycle, which is clearly different from the noise observed in some of our traces of unidirectionally aligned dyneins.

**Author response image 1. sa2fig1:** Comparison of the noise before and after UV flash.

Reviewer #2 (Recommendations for the authors):The authors could discuss how the force generating dyneins are localized in cilia. In Figure 8c, dynein generates force at the straight parts, not at the bending point. This is close to the model based on the sliding disintegration (Hayashi and Shingyoji, 2008) and different from the one proposed based on cyo-ET (Lin and Nicastro), where asymmetrical dynein activation occurs at the bending points.

We agree with the reviewer that distribution of the force-producing dyneins is an interesting and important issue. However, our results do not reveal the localization of force-producing dynein, and therefore we feel that it would be over-interpretation to discuss it.

Reviewer #3 (Recommendations for the authors):1. Please explain in the Abstract that the paired microtubules are arranged in parallel, otherwise it is hard to understand that 'opposite orientation' (of the dynein sets) refers to their originating on different microtubules.

Thank you for the comment. We have revised the sentences in Abstract as the following:

“Electron microscopy (EM) showed pairs of parallel MTs crossbridged by patches of regularly arranged dynein molecules bound in two different orientations depending on which of the MTs their tails bind to. The oppositely oriented dyneins are expected to produce opposing forces when the pair of MTs have the same polarity.”

2. I don't understand why restraining force due to the DNA crosslinker triggers oscillatory switching between the two sets of dyneins, whereas restraining force due to the optical trap does not. I may be missing something, please comment?

Thank you for pointing out an interesting possibility. We think that the optical trap force can basically trigger oscillation. However, we used small beads (200 nm) in order to detect 8-nm steps, so that the trap force was weak. As the results, the optical trap could not stop sliding of the MTs, and the dynein-MT complexes disassembled in the absence of the DNA linkers.

When a larger bead (500 nm) was used to measure the maximum force, the optical trap was able to stop sliding of the MTs, and some traces showed up and down at a higher force (Figure 3 – supp 1). By adjusting the conditions such as the number of dyneins per complex, it might be possible to see oscillatory switching of the moving direction by the optical trap force.

3. I don't understand why in Figure 5B, the initial response to the UV flash is 170 nm of unidirectional sliding, after which much smaller amplitude oscillations ensue. Were the DNA linkers not initially engaged? Can the dynein drag the origami? According to Figure 4 supp1, 96 nm is the max possible sliding distance. Again I may be missing something. There are similar large amplitude sliding events in some of the traces in Figure 5 supp 1.

A similar comment was made by Reviewer #1 (Recommendations for the authors #6). As this reviewer suggests, we think that the dyneins can drag origami when not enough origami rods are functioning. As described in Materials and methods, the end of a linker of the DNA origami rod has three sites for mutant kinesin heads to attach. Thus, the origami is bound to a MT via one to three kinesin heads. Although there are several origami rods crosslinking the MTs, they are in various angles, and when the MTs slide relative to each other, one of the linkers would become fully stretched to maximum before the other linkers do. We think that the binding force of one or two kinesins may not be strong enough, and they may be dragged until cooperation of some more linkers stops the sliding.

4. l.186 How are oscillation events defined?

Thank you very much for the comment. Although we selected the traces in which we could detect steps for the analysis in Figure 6, we did not have a clear definition for this sentence. To answer this comment, we have now set a definition, re-counted the traces, and modified the text:

Lines 208-211: “Forty-eight out of 94 such traces showed sliding movement, and 65% of them exhibited oscillatory movement (at least two forwards and two backwards displacements, each displacement larger than 10 nm, and the velocity of each movement between 0.1 µm/s and 50 µm/s) in some parts of the trace.”

We chose the value ‘50 µm/s’ because sliding velocities faster than 40 µm/s were observed for dynein-MT-complexes by Aoyama and Kamiya (Cytoskeleton, 2010).

5. Does the steppiness of the traces imply that relatively few dyneins (perhaps sometimes only one?) are engaged at any one time? Or are multiple dyneins synchronising? In Fig7 Supp2 (no origami, larger forces) are the traces steppy?

The average number of dyneins per complex was 35. If the numbers of oppositely oriented dyneins are equal, the number of the dyneins oriented in the same way would be ~18. Because of the low duty ratio of outer-arm dynein, it is likely that the number of dyneins engaged at any one time is small.

Prompted by this comment, we tried step analysis for the traces in Figure 7 and Figure 7-supp2, and steps were indeed detected for some of them. An example (a part of the trace in Figure 7-supp.2C) is shown in Author response image 2.

**Author response image 2. sa2fig2:** Steps detected for the trace in Figure 7 —figure supplement 2C.

6. Concerning the possible difference in the velocity distribution for backwards sliding versus forwards sliding (relative to the trap centre) (Figure 6C), the authors consider the possibility that the trapping potential may be hindering the one and helping the other, and raise the possibility that some backwards motions are slips. Can these events not be recognised according to their special rapidity? If so, are there any 'forwards' detachment events?

With the step-finding algorithm we used, we estimated by simulation that we can reliably (>90 %) detect 8-nm displacement only when the dwell time is longer than 1 ms. Therefore, we could not tell whether the very rapid movements included steps or they were just ‘slips’.

We did not recognize any clear ‘forwards detachment’. During forwards (away from the trap center) movement, the trap force, the force of the DNA linkers, and that of the oppositely oriented dyneins are all directed to the trap center, and the only force directing away from the trap center is produced by the dyneins that are moving forwards. Therefore, it is most likely that ‘detachment’ occurs in the backwards direction.